# Unraveling the Link between Ιnsulin Resistance and Bronchial Asthma

**DOI:** 10.3390/biomedicines12020437

**Published:** 2024-02-16

**Authors:** Konstantinos Bartziokas, Andriana I. Papaioannou, Fotios Drakopanagiotakis, Evanthia Gouveri, Nikolaos Papanas, Paschalis Steiropoulos

**Affiliations:** 1Private Pulmonologist, 42132 Trikala, Greece; bartziokas@gmail.com; 21st University Department of Respiratory Medicine, “Sotiria” Hospital, National and Kapodistrian University of Athens, 15772 Athens, Greece; papaioannouandrianna@gmail.com; 3Department of Pneumonology, Medical School, Democritus University of Thrace, 68100 Alexandroupolis, Greece; fdrakopanagiotakis@gmail.com; 4Diabetes Centre, 2nd Department of Internal Medicine, Democritus University of Thrace, 68100 Alexandroupolis, Greece; vangouv@yahoo.gr (E.G.); papanasnikos@yahoo.gr (N.P.)

**Keywords:** asthma, insulin resistance, antidiabetic drugs, metabolic dysfunction, obesity

## Abstract

Evidence from large epidemiological studies has shown that obesity may predispose to increased Th2 inflammation and increase the odds of developing asthma. On the other hand, there is growing evidence suggesting that metabolic dysregulation that occurs with obesity, and more specifically hyperglycemia and insulin resistance, may modify immune cell function and in some degree systemic inflammation. Insulin resistance seldom occurs on its own, and in most cases constitutes a clinical component of metabolic syndrome, along with central obesity and dyslipidemia. Despite that, in some cases, hyperinsulinemia associated with insulin resistance has proven to be a stronger risk factor than body mass in developing asthma. This finding has been supported by recent experimental studies showing that insulin resistance may contribute to airway remodeling, promotion of airway smooth muscle (ASM) contractility and proliferation, increase of airway hyper-responsiveness and release of pro-inflammatory mediators from adipose tissue. All these effects indicate the potential impact of hyperinsulinemia on airway structure and function, suggesting the presence of a specific asthma phenotype with insulin resistance. Epidemiologic studies have found that individuals with severe and uncontrolled asthma have a higher prevalence of glycemic dysfunction, whereas longitudinal studies have linked glycemic dysfunction to an increased risk of asthma exacerbations. Since the components of metabolic syndrome interact with one another so much, it is challenging to identify each one’s specific role in asthma. This is why, over the last decade, additional studies have been conducted to determine whether treatment of type 2 diabetes mellitus affects comorbid asthma as shown by the incidence of asthma, asthma control and asthma-related exacerbations. The purpose of this review is to present the mechanism of action, and existing preclinical and clinical data, regarding the effect of insulin resistance in asthma.

## 1. Introduction

Asthma is one of the most common respiratory disorders worldwide [1], and in 2019, there were 262.4 million prevalent cases and 37.0 million new cases of asthma all over the world, contributing to 21.6 million disability-adjusted life years (DALYs) [2]. Chronic eosinophilic bronchitis with T helper 2 (Th2) inflammation, bronchial smooth muscle hyperplasia, easy epithelium shedding, mucus plug formation in the bronchi, and thickening of the subepithelial basement membrane are its hallmark pathologic features [1]. Due to its broad diagnostic standards, which include varying expiratory airflow limitation and respiratory symptoms, asthma is a heterogeneous disease with multiple pathogenic mechanisms (endotypes) and higher-order groupings sharing similar clinical characteristics (phenotypes) [3,4]. In the era of personalized medicine, significant progress has been achieved in disease phenotyping and endotyping with the evolution and utilization of dependable biomarkers precisely measured through reliable and repeatable techniques [5]. Even though several asthma phenotypes have been described, including allergic and non-allergic asthma, occupational, environmental (indoor/outdoor pollutants) [6] and aspirin-induced asthma, the pathologic characteristics of these phenotypes are very similar, while type 2 inflammation by Th2 and group 2 innate lymphoid (ILC2) cells is their crucial component [3]. Regarding severe uncontrolled asthma phenotypes, recently developed monoclonal antibody treatments have highlighted the importance of type 2 inflammation [7]. Despite the differences in the design of different studies, it is clear that a considerable minority of patients with severe uncontrolled asthma have a lack of atopy and type 2 inflammation, emphasizing the significance of non-allergic triggers to these patients [8,9].

Dysglycemia, glycemic dysfunction, and glucose dysmetabolism, (also referred to as disorders of glucose metabolism) have been identified as possible causes of severe asthma and as potential triggers of asthma exacerbation [10,11]. Asthma and insulin resistance appear to interact and interrelate through mechanisms and pathways that are not yet fully understood, thereby influencing each other’s natural course [12]. Compared to individuals with Th2 inflammation, patients with asthma and metabolic dysfunction have additional pathogenic and pathologic characteristics [11]. Often presented as metabolic syndrome, metabolic dysfunction has several significant clinical characteristics, such as insulin resistance/glucose intolerance, central obesity, dyslipidemia, and in certain cases, vitamin D (Vit D) deficiency [13,14].

It is commonly known that the treatment of asthma in obese patients can be challenging, partly because of the patients’ poor response to inhaled corticosteroids [15,16]. Treatment of insulin resistance has favorable effects on asthma management [17]. Additionally, metabolic dysfunction appears to be the primary cause of asthma exacerbation, according to a cohort analysis of asthmatic patients who were prone to exacerbations [18]. These characteristics highlight the association between asthma phenotype and metabolic dysfunction, and the significance of metabolic dysfunction in the pathogenesis of asthma. The impact of obesity and metabolic dysfunction on asthma is becoming increasingly apparent as these diseases’ prevalence rises worldwide.

In this context, the purpose of this review was to summarize current data on the deleterious role of insulin resistance in asthma, along with suggested mechanisms of asthma irritation.

## 2. Insulin Resistance

The most comprehensive definition of “insulin resistance” was formulated by Petersen and Shulman [19] as a “maladaptive in the setting of chronic overnutrition”. Even though its molecular causes are not entirely known, improper lipid accumulation in the liver is considered a major factor that contributes to decreased cellular sensitivity to insulin. To preserve glucose homeostasis, pancreatic B-cells produce more insulin, which results in a period of clinically silent hyperinsulinemia with normal blood glucose levels [10].

With time, blood glucose levels rise in tandem with reductions in insulin secretion as this compensatory hyperinsulinemia starts to falter [20]. This period can be identified by elevated glycated hemoglobin A1c (HbA1c), which is a measure of the average blood glucose levels over the previous 3 months, elevated fasting glucose (also named impaired fasting glucose, or IFG), or inadequate ability to completely offset hyperglycemic increases as a result of a glucose load (also named impaired glucose tolerance or IGT). Elevated IFG, IGT, or HbA1c are useful in diagnosing prediabetes [21]. Crucially, the human body still secretes more insulin during the prediabetic state, and eventually, there is a reduction in insulin secretion that coincides with overt hyperglycemia and pancreatic failure. Diabetes is diagnosed based on blood glucose or HbA1c levels [22].

Evidence from the Third National Health and Nutrition Examination Survey (NHANES III) has shown that one-third of nondiabetic adults in the United States (US) experience hyperinsulinemia associated with insulin resistance [23], whereas 12–43% of US adults have prediabetes. In the US, 11.3% of the population of all ages (more than 37 million people), had diabetes in 2019 [24]. Taken together, these data suggest that insulin resistance, prediabetes and diabetes are widespread and interrelated. Metabolic syndrome, which includes obesity, hypertension, dyslipidemia, and hyperglycemic tendencies, is closely linked to obesity (specifically abdominal obesity), while insulin resistance and hyperinsulinemia are the main molecular markers of this syndrome [25]. Indeed, large epidemiological studies [26,27] have shown that the main obesity-associated asthma risk was attributed to insulin resistance, and even maternal obesity raises the incidence of childhood asthma in susceptible children, pointing to systemic causes [28]. In the Korean Health and Genome Study (a population-based study of 10,038 Korean adults aged 40 to 69) [29], metabolic syndrome was related to asthma-like symptoms.

There is a strong connection between more severe and uncontrolled asthma and metabolic syndrome [30], which are often related to increased airway hyperresponsiveness, oxidative stress, regulation of systemic and pulmonary inflammation, and macrophage activation [31,32,33]. Modifications to nutrition and microbiome, immunity, inflammation and biophysical lung function associated with obesity may also likely have an impact on asthma, regardless of metabolic pathways [14,34]. Particularly, obesity-induced insulin resistance seems to precede macrophage accumulation and inflammation in adipose tissue [35] and has been linked to elevated inflammatory markers in serum and adipose tissue among patients with severe asthma [34]. On the other hand, sputum and serum expression of interleukin 6, which is recognized to participate in the development of insulin resistance, have been linked with more severe asthma and airway obstruction [36,37].

## 3. Insulin Resistance and Asthma Pathophysiology: A Missing Link

The degree of insulin resistance and the amount of circulating insulin are strongly correlated in most patients with prediabetes and insulin resistance. As a result, the lung and other peripheral organs that retain insulin sensitivity are susceptible to high levels of circulating insulin [30]. Studies have shown the presence of insulin receptors in the developing lung [38], but while the lungs’ extent of insulin sensitivity is unknown, it is assumed that (like most other tissues) hyperinsulinemia is accompanied by reduced insulin-like growth factor binding proteins (IGFBP-1 and -3) and increased free insulin-like growth factor (IGF-1) [39,40]. Both insulin and IGF-1 have significant impacts on cell proliferation and differentiation through several mechanisms of action, including increased fibrosis, increased epithelium to mucus transition, as well as increased contractility and mass of airway smooth muscle (ASM); some of these are associated with asthma phenotypes.

Insulin causes a hypercontractile phenotype of bovine ASM, similar to that seen in asthma, by increasing the production of b1-containing laminins through a phosphoinositide-3 kinase (PI-3K)/Akt-dependent signaling pathway [41]. Recent observations in humans [41] have shown that inhaled insulin can cause an abrupt loss in lung function due to airway smooth muscle contraction, suggesting that hyperinsulinemia may increase airway smooth muscle bulk or contraction. In mice, administration of intranasal insulin resulted in an enhancement in the deposition of collagen in the lungs, as well as increased airway hyper-responsiveness [42]. Furthermore, the lungs of mice treated with insulin showed activation of β-catenin by PI3K/Akt, which is a positive regulator of epithelial-mesenchymal transition and fibrosis, suggesting that hyperinsulinemia may have negative impacts on airway structure and function.

A key mechanism implicated in several biological processes, such as cell proliferation, morphogenesis, and development, is the Wnt/β-catenin-signaling pathway [43]. Multiple human abnormalities, such as malignancies and inflammatory, fibrotic, and metabolic disorders, have been linked to aberrant Wnt/β-catenin signaling [44]. Experimental studies in a murine model have illustrated that blocking Wnt/β-catenin signaling decreases pulmonary fibrosis [45]. The wnt/β-catenin-signaling pathway plays a crucial role in pulmonary arterial hypertension, in which vascular smooth muscle cell proliferation is a fundamental feature [46]. Considering that smooth muscle hyperplasia and subepithelial fibrosis are the hallmarks of airway remodeling, we can speculate that Wnt/β-catenin signaling plays a role in the process of airway remodeling in asthma. In the lung tissue of mice with chronic asthma, the use of a particular siRNA, which blocks β-catenin expression, resulted in airway remodeling and inflammation, reduction of subepithelial fibrosis and collagen accumulation, and downregulation of transforming growth factor-β production [47]. Furthermore, suppression of β-catenin in a model of chronic asthma prevented smooth muscle hyperplasia through downregulation of the tenascin C/platelet-derived growth factor receptor pathway, indicating that this pathway is abundantly expressed and controls the process of airway remodeling. Hence, insulin is thought to be involved in b-catenin-mediated unfavorable airway remodeling, since inhibiting this protein has been demonstrated to block the progression of subepithelial fibrosis, mucus metaplasia, and ASM hyperplasia in chronic asthma [48].

Remodeling of the airway wall is a feature of chronic airway inflammation and may play a significant role in airway hyperresponsiveness and lung function loss in patients with severe persistent asthma [49]. Both changes in extracellular matrix (ECM) [50] and increased mass of airway smooth muscle (ASM) [51] are characteristics of airway wall remodeling in these patients. ECM proteins known as laminins are frequently present in basement membranes with an increased expression being observed in the airways of patients with asthma compared with healthy controls [52]. Using bovine tracheal smooth muscle, Dekkers et al. showed a pivotal role for laminins in the establishment of an ASM phenotype that is hypercontractile and hypoproliferative after insulin exposure [41]. Thus, increased ASM contractility and contractile protein expression may be related to increased laminin synthesis by ASM in asthma. Moreover, the same study showed that eight days of administration of insulin causes an elevation in the expression of particular markers of the contractile phenotype in the smooth muscle cells and strips of the bovine trachea [53]. This was followed by a decrease in mitogenic responses and the installation of a phenotype functionally hypercontractile [54].

Additionally, it has been observed that high-fat-diet (HFD)-induced obesity enhanced the expression of TGF-β1 and insulin resistance in the lungs, which causes perivascular and peribronchial pulmonary fibrosis and aggravated airway hyper-responsiveness (AHR) to inhaled aerosolized methacholine (MCh) in mice [55].

Intranasal insulin increased the expression of TGF-β1 in the bronchial epithelium and caused lung fibrosis. HFD-induced AHR, lung fibrosis, and goblet cell hyperplasia were reduced by the anti-TGF-β1 antibody. Regarding airway hyper-responsiveness, muscarinic M2 receptors in parasympathetic nerves in the trachea of humans and rats are inhibited by insulin, which leads to an increase in acetylcholine release and airway contraction [56]. Loss of inhibitory M2 muscarinic receptor function on parasympathetic nerves and enhanced vagally mediated bronchoconstriction in obesity are strongly related [56]. Interestingly, Calco GN et al. demonstrated that cutting insulin receptors on sensory nerves blocked the increase in sensory nerve density and inhibited airway hyperreactivity in obese mice with hyperinsulinemia [57]. The same author [58], in a mouse model of maternal diet-induced obesity that recapitulates metabolic dysregulation, showed for the first time, that exposure to a maternal high-fat diet leads to hyperinnervation of airway sensory nerves and increased reflex bronchoconstriction in offspring fed a regular diet only [58]. These findings could clarify why obese people are more likely to experience asthma exacerbations since hyperinsulinemia is more common and predominant in obese people. They also imply that anticholinergic medications may be useful in treating this type of asthma. Furthermore, insulin-exposed open-ring guinea pig tracheal preparations induce ASM contraction via PI3-kinase and Rho kinase-dependent mechanisms mediated by contractile prostaglandin synthesis [59]. Xu et al. found [60] that insulin treatment decreased the β-agonist responsiveness of primary human ASM cells and obese mice via phosphorylating phosphodiesterase 4D and upregulating its downstream activity. In contrast, a recent study by Ferreira et al. [61] demonstrated that insulin deficiency correlated with decreased lung levels of STAT3, JNK, and ERK1/2. Additionally, in a mouse model of asthma, it blocked the onset of allergic inflammation, eosinophilic pulmonary infiltration, and airway hyperresponsiveness [61].

Delving even deeper into the pathophysiology of asthma, it is worth mentioning that mast cells are essential for the development and progression of inflammatory and acute-type allergic reactions. Lessman et al. [62], found that in mast cells generated from rabbit bone marrow, insulin and IGF-1 increased cell survival through the PI3-kinase pathway. These effects might be involved in the inverse link between atopic diseases, such as allergies and asthma, and type 1 diabetes mellitus, which is characterized by low insulin levels. Insulin-mediated stimulation of PI3-kinase and ERK pathways prevented human bronchial epithelial cell apoptosis, which may facilitate airway remodeling [63]. Accordingly, insulin is important in the pathogenesis of apoptosis-driven lung diseases (such as asthma and chronic obstructive pulmonary disease).

The lungs are not sterile, but they have a far lower bacterial load compared with other mucosal surfaces. This is partly because of strict control over the availability of nutrients, including glucose. Normally, the airway surface liquid has up to 12 times lower glucose levels compared with plasma [64]. Two mechanisms maintain this low concentration: glucose transporters that take up the glucose from the blood, and tight junctions that block it from entering the airway. Keeping glucose levels low may be a homeostatic strategy that prevents bacteria from growing by depriving them of an essential nutrient [65]. A higher risk of bacterial lung infection arises when these processes fail with resultant glucose increase, particularly in patients with underlying lung disease, like asthma [66]. In line with this, high glucose levels increase the likelihood of respiratory tract bacterial colonization in critically ill patients [66].

In the event of chronic hyperglycemia, advanced glycation end-products (AGEs) are actively generated and accumulate in the bloodstream and different organs [67]. By various mechanisms, AGEs also promote the expression of AGE receptors and play a crucial role in the onset of vascular complications in diabetes. Initially, the multiligand receptor known as the receptor for advanced glycation end products (RAGE) was suggested as a potential mediator in diabetes [68]. However, it was later discovered that membrane RAGE (mRAGE) signaling is pro-inflammatory, while soluble RAGE (sRAGE), a secreted form of RAGE, is mostly anti-inflammatory due to its ability to scavenge pro-inflammatory ligands [68]. Milutinovic et al. [69], revealed that RAGE is essential to the disease processes that lead to pulmonary eosinophilia, mucus hypersecretion, airway remodeling, and hyper-responsiveness of the airways in a model of dust mite-induced asthma/allergic airway disease. Airway hypersensitivity (resistance, tissue damping, and elastance), eosinophilic inflammation, and airway remodeling were all eliminated in the absence of RAGE. Additionally, the expression of IL-5 and IL-13 protein and mRNA in the lung was decreased [69].

Finally, insulin resistance, hyperglycemia, or both have been linked as a cause of accelerated decline in respiratory function, a condition that has been reported as “diabetic lung” [70]. The term “diabetic lung” includes several abnormalities of the respiratory function concerning control of ventilation, bronchomotor tone, lung volume, pulmonary diffusing capacity, and neuroadrenergic bronchial innervation. Although the precise pathogenetic processes through which insulin resistance, hyperglycemia, and diabetes mellitus may affect respiratory function are not yet fully understood, it is speculated that an increase in lung collagen and elastin [71], and the occurrence of subclinical nodular fibrosis, in addition to physiological impacts on the function of the respiratory muscles are responsible for this effect [72].

The suggested mechanisms of insulin resistance in asthma are shown in Table 1 and Figure 1.

## 4. Epidemiological-Observational Studies

Diabetes and disorders of glucose metabolism have been associated with reduced diffusing capacity and restrictive spirometry in subpopulations free from any pulmonary disease. This is based on 40 clinical trials evaluating lung function data of 3182 patients with diabetes [73]. However, no study has clarified whether hyperglycemia and consequently diabetes mellitus, increases the risk of asthma or vice versa [74]. It appears to be a bidirectional independent relationship between diabetes mellitus and asthma [74].

One of the first studies evaluating this issue was conducted in Denmark [26] and concluded that insulin resistance was related to a higher probability of manifesting asthma-like symptoms, thus promoting the theory that asthma and obesity may be associated with inflammatory processes also implicated in insulin resistance.

A few years afterward, Mueller et al. [75], using data from Singapore Chinese Health Study, also found a positive correlation between the likelihood of developing type 2 diabetes and self-reported, physician-diagnosed asthma. In agreement with the two aforementioned studies, the Nord-Trøndelag Health Study (HUNT) examining prospectively 23191 adults aged 19–55 years, without asthma at baseline, showed that metabolic syndrome was related to increased risk of incident asthma, indicating that physicians may take metabolic syndrome into account as a predictor of future risk of asthma [76]. The same was the result from another study that used multiple logistic regression analyses from the National Health and Nutrition Examination Survey (NHANES), 2001–2018, and found that an increase in the metabolism score for visceral fat (METS-VF) index was related to an increase in the incidence of asthma [77]. Recently, a cross-sectional analysis of 41,480 adults from the Korean National Health and Nutrition Examination Survey during the period 2007–2016, revealed that when both obesity and metabolic syndrome are present, the risk of developing asthma is highest [78]. The above observation is also seen in children: a study from Australia Illustrated signs of insulin resistance in 43% of children aged 6 to 17 who suffered from allergic asthma [79].

Together with the aforementioned epidemiological population-based studies, studies employing electronic and administrative health records similarly reported that the probability of developing asthma is higher in subjects with type 2 diabetes compared with non-diabetic individuals, suggesting that these 2 diseases are related [80,81]. Nevertheless, a major limitation in all these studies is that incident asthma was defined by self-report or through diagnostic codes and not objectively confirmed.

One step forward, Cardet et al. [82] hypothesized that insulin resistance is an effect modifier of the relationship between asthma and obesity. A history of physician-diagnosed current asthma and insulin resistance was obtained from 12,421 individuals, aged 18–85 years, utilizing the large National Health and Nutrition Examination Survey from 2003–2012. In logistic regression analysis, increased insulin resistance increases in obese individuals increased the likelihood of asthma [82]. This was not the case for other components of metabolic syndrome [82]. Therefore, it is possible that a subgroup of patients with asthma and obesity may be identified by their insulin resistance, and agents targeting insulin resistance may also improve asthma control in this subgroup [82]. This was confirmed recently, in another cross-sectional study of 1276 participants from the NHANES 2009–2012 database [83]. In this analysis, waist-to-hip ratio and insulin resistance independently predicted impaired pulmonary function in overweight/obese asthmatic adults [83]. Evaluations were based on forced expiratory volume in the 1st second (FEV_1_), forced vital capacity (FVC), and forced expiratory flow over the middle half of the FVC (FEF25–75_%_).

As indicated in a recent study by Peters and colleagues [84], in a SARP-3 (Severe Asthma Research Program–3) cohort, insulin resistance is independently linked to airflow limitation, diminished treatment responses, and an accelerated deterioration in lung function over time. In more detail, patients with insulin resistance had considerably lower FEV_1_ and FVC values and responded to β-adrenergic agonists and systemic corticosteroids with a lower FEV_1_ compared to individuals without insulin resistance. Moreover, individuals with moderate and severe insulin resistance had an annualized decline in FEV_1_ of −41 mL/year and −32 mL/year, respectively, compared to patients without insulin resistance who had an annualized decline of −13 mL/year [84]. What is noteworthy about the present study is the fact that after controlling for body mass index in regression models, the analysis showed that the effects of extra body fat on chest wall mechanics were unlikely to fully explain the association between worsening lung function and insulin resistance [84,85].

Accordingly, the association between insulin resistance, diabetes, and prevalent asthma is bidirectional. Other studies including patients with asthma have also reported a positive correlation between disorders of glucose metabolism and asthma morbidity. An increased risk of asthma exacerbation has been associated with a higher incidence of diabetes, hypertension, and obesity, according to a recent longitudinal analysis of the Severe Asthma Research Program cohort [18]. In a cross-sectional analysis using data from ERICA (Study of Cardiovascular Risk in Adolescents, Portuguese acronym ERICA), a multicenter, school-based countrywide study in a complex sample of adolescents aged 12–17 years, metabolic syndrome and insulin resistance were significantly associated with severe asthma in adolescents from Brazilia [86].

In an attempt to ascertain the relationship between pre-diabetes/diabetes and asthma exacerbations in an obese asthma cohort, Wu et al. [87], conducted a retrospective cohort of 5722 individuals with obese asthma in the United States, aged 18–64, from a claims-based health care database covering 2010–2015. It is worth underlining that in the current study, the investigators used HbA1c instead of a historical or self-reported diagnosis of diabetes, and pre-diabetes was defined as 5.7% ≤ HbA1c ≤ 6.4%, while diabetes was defined as HbA1c ≥ 6.5% [87]. Compared to patients with normal HbA1c, those in the pre-diabetes range experienced a 27% and those in the diabetes range experienced a 33% higher asthma exacerbation rate. Yang et al. [17] conducted a cross-sectional analysis of 47,606 adults with asthma who did not have diabetes mellitus from the UK Biobank to evaluate the association between lung function, HbA1c, and asthma-related hospitalizations. Both HbA1c per se and an HbA1c in the pre-diabetic or diabetic range were associated with ≥1 asthma hospitalization. Both HbA1c per se and HbA1c in the prediabetic/diabetic range were significantly and inversely associated with FEV_1_ and FVC [17].

Other studies have also shown that prediabetes or diabetes is linked with asthma exacerbations [88,89]. Although non-significant after adjustment for covariates, diabetes has been associated with asthma exacerbations [88]. A cross-sectional analysis of data from 709 participants in the US-based Severe Asthma Research Program (SARP)-3 cohort revealed that patients with exacerbation-prone asthma were more likely to self-report a diagnosis of diabetes mellitus than those without exacerbation-prone asthma [88]. Conversely, even after controlling for smoking status, overweight or obesity, and other potential confounders, a study of 130,547 patients (aged 12–80 years) in two UK databases found that type 1 or type 2 diabetes was strongly related to 1.53 times increased odds of hospitalizations due to asthma within the following year [89].

Finally, to evaluate the relationship between triglyceride-glucose index (TyG), a biomarker of metabolic syndrome and insulin resistance, with the risk of severe asthma exacerbation, Staggers KA et al. [90] designed a 5-year retrospective study. In it 108,219 patients with asthma from the US Veterans Health Administration were followed for a severe asthma exacerbation, from January 2015 to December 2019. The authors found that independent of obesity, eosinophils, smoking and asthma treatment, patients with an elevated TyG had a 6% greater risk of experiencing a severe asthma exacerbation [90].

Insulin resistance and glucose dysregulation have been further associated with impaired lung function in people with and without asthma. A multivariable analysis of 15,792 United States adults in the Atherosclerosis Risk in Communities (ARIC) Study showed that the % predicted of FEV_1_ and FVC were 2.4% and 3.6% lower, respectively, in adults with diabetes compared to those without diabetes after adjusting for smoking, body-mass index, waist circumference, and other covariates [91]. Conversely, in a U.S. nationwide survey study of 4257 adults without diabetes, there was a significant non-linear inverse association between elevated HbA1c and FEV_1_, FVC, and FEV_1_/FVC ratio, after adjusting for body-mass index and waist-to-hip ratio [92]. Unfortunately, the possible effects of respiratory comorbidities, including asthma, on glucose metabolism were not taken into account in either of those two trials. Moreover, according to a cross-sectional study of 1429 adolescents between the ages of 12 and 17 from the United States who participated in the 2007–2010 National Health and Nutrition Examination Survey [93], metabolic syndrome and insulin resistance were linked to impaired lung function in overweight/obese adolescents. In another cross-sectional study of diabetic and non-diabetic adults [94], a 1% absolute increase in HbA1c was associated with a −52 mL difference in FVC and a −25 mL difference in FEV_1_ in women, and a −128 mL difference in FVC and a −73 mL difference in FEV_1_ in men, showing an inverse association between glycemic measurements and pulmonary function. The findings of an additional cross-sectional observational cohort study of non-smoker African American adults from the Jackson Heart Study were also similar [95]. There was no significant difference in lung function between women and men with impaired glucose tolerance and those with normal glucose tolerance, while women and men with diabetes had lower FEV_1_ and FVC than those with normal glucose tolerance [95].

Ultimately, data from the Colombian Diabetes Association Center in Bogotá [96], showed that in comparison to patients with adequate control, patients with type 2 diabetes and inadequate glucose control had higher FEV_1_/FVC (0.013%) and lower mean residuals of FEV_1_ (75.4 mL) and FVC (121 mL), as well as higher levels of all inflammatory markers.

In light of all these data, we can assume that insulin resistance and metabolic syndrome are strongly related to asthma prevalence and may predict the impairment of lung function. However, the underlying mechanisms remain unclear. In an attempt to answer this query, 168 Hispanic and African American adolescents (13–18 years) from Children’s Hospital in Montefiore were divide.d into groups of 42 obese subjects with asthma, 42 normal-weight subjects with asthma, 40 obese subjects without asthma, and 44 healthy control subjects [33]. In children with obesity-related asthma, lung function deficiencies and non-atopic systemic inflammation have been linked to insulin resistance and dyslipidemia [33].

## 5. Current Antidiabetic Drugs and Asthma

If diabetes and insulin resistance aggravate asthma, it makes sense that treating these conditions could have a positive therapeutic impact, provided that the mechanism causing lung damage is reversible. In addition to their effects on glucose management, many antidiabetic drugs have additional actions that can also affect asthma [97].

According to in vivo and in vitro studies of the last decade, metformin exerts anti-inflammatory effects in airways [98,99]. In mice asthma models, metformin has been demonstrated to reverse eosinophilic infiltration of the lung tissue and reduce levels of pro-inflammatory cytokines, reactive oxygen species, and nitric oxide species [99]. Adenosine monophosphate-activated protein kinase (AMPK) is thought to be the likely key mechanism by which metformin has an anti-inflammatory effect on the airway [99]. Metformin has been observed to activate AMPK, and in a dose-dependent manner inhibit tumor necrosis factor (TNF)-α–induced NF-κB activation and TNF-α–induced IκB kinase activity [100]. Additionally, this agent appears to attenuate the TNF-α–induced gene expression of various proinflammatory and cell adhesion molecules (such as vascular cell adhesion molecule-1, E-selectin, intercellular adhesion molecule-1, and monocyte chemoattractant protein-1) in human umbilical vein endothelial cells (HUVECs) [100]. AMPK activity also reduces oxidative stress and contributes to the inhibition of tumor necrosis factor-α-induced inflammatory signaling and expression of inducible nitric oxide synthase mediated by nuclear factor-kB [99]. In male mice given a high-fat diet (HFD) for ten weeks to induce obesity, Calixto et al. showed that metformin reduced the exacerbation of allergic eosinophilic inflammation [98]. Given that metformin is proposed as a first-line therapeutic medication for diabetes [101], it may be a new therapeutic option for these patients. Indeed, metformin was associated with a reduced risk of asthma-related emergency department visits and hospitalizations in a US cohort of patients with diabetes and asthma [102].

An 11-year (2001–2011) retrospective cohort study that followed 1332 individuals with concurrent diabetes and asthma for three years to evaluate the asthma-related outcomes was conducted utilizing the Taiwan National Health Insurance Research Database [103]. The study concluded that metformin users experienced a lower risk of asthma exacerbation or asthma-related hospitalization compared to non-users [103]. Very recently, Wu et al. [104] identified 1749 patients with asthma and diabetes using the Johns Hopkins electronic health record from April 2013 to May 2018. The use of metformin, independently of glycemic control and obesity, was related to a lower hazard of asthma-related emergency department visits and hospitalizations [104]. Unfortunately, there are no data regarding the beneficial role of metformin from prospective controlled studies in asthmatic patients, with and without obesity.

Glucagon-like peptide 1 (GLP-1) is a hormone derived from the gut that enhances insulin sensitivity and increases pancreatic insulin production in response to oral food consumption [105]. The GLP-1Ra (GLP-1 receptor) exists in the lung and immune cells and may mediate several inflammatory pathways implicated in the pathogenesis of obesity-related asthma [106]. Preclinical studies have indicated that GLP-1 inhibited the production of pro-inflammatory cytokines, including TNF-a, by inactivating NF-κB in a protein kinase A-dependent manner [107,108]. Moreover, GLP-1 receptor agonists (GLP-1RAs) suppress RAGE expression, and thus reduce inflammation and bronchoconstriction [109]. Furthermore, the GLP-1RA liraglutide reduced mucus hypersecretion and airway inflammation in a mouse model of allergic asthma [110]. In addition, in mice models, GLP-1RAs reduced the release of T2 cytokines from type 2 innate lymphoid cells (ILC2), as well as mucus production after exposure to viral antigens and fungal allergens [111,112,113]. Interestingly, in isolated human airways, GLP-1R activation reduced contractile tone and reduced lipopolysaccharide-stimulated eosinophil activation [114,115].

GLP-1RAs have been linked with improvements in asthma outcomes in patients with asthma and diabetes mellitus. In the first uncontrolled preliminary study, 9 participants receiving a GLP-1RA for one-year improved asthma symptoms and decreased asthma exacerbations [116]. An electronic health records-based new user, active-comparator, retrospective cohort study of patients with type 2 diabetes and asthma showed that patients on GLP-1RAs for type 2 diabetes had fewer asthma exacerbations compared with other antidiabetic agents (sodium-glucose cotransporter-2 inhibitors [SGLT-2is], dipeptidyl peptidase inhibitors [DPP-4is], sulfonylureas, or basal insulin) [117].

Another new-user active-comparator analysis using a national claims database (2005–2017) also showed that patients with diabetes and chronic lower respiratory disease (CLRD) (a medical term that includes both COPD and asthma) who started GLP-1RA experienced fewer CLRD exacerbations compared to those starting DPP-4is [118]. In diabetes, a randomized clinical trial reported that liraglutide decreased serum surfactant protein D, which independently predicted improvements in FVC [119]. The addition of a GLP-1RA to metformin, as opposed to metformin alone or metformin plus insulin, improved lung function (FEV_1_ and FVC) in a prospective cohort of 32 patients with diabetes who did not have obstructive lung disease [120].

Sulfonylureas are frequently used as second-line antidiabetic therapy [121]. Sulfonylureas attach to receptors on the cell membranes of pancreatic beta cells and increase insulin secretion. Their main untoward effect is hypoglycemia [122,123]. The utilization of a representative UK primary care database in a retrospective cohort study revealed an association between sulfonylureas and a lower risk of incident asthma [124].

DPP-4is are preferable to sulfonylureas as second-line therapy after metformin initiation, due to their neutral effect on weight and the absence of hypoglycemia [125,126,127]. Very recently, studies utilizing in vitro models of human bronchial epithelial cells have demonstrated that DPP-4i inhibits pathways that lead to fibrosis [128] and oxidative stress [129]. Nonetheless, a retrospective observational matched cohort study found that treatment with DPP-4i did not improve asthma control, treatment stability or asthma exacerbations [130]. This has been confirmed in a network meta-analysis [131].

SGLT-2is promotes glycosuria [132]. An in vitro transcriptomics experiment in human proximal tubular cells revealed that the SGLT-2i canagliflozin decreased TNF receptor 1, matrix metalloproteinase 7, IL-6, and fibronectin 1 during two years of follow-up, as compared with the sulfonylurea glimepiride [133].

In 2021, 9 large randomized controlled trials (RCTs) were included in a fixed-effects meta-analysis to assess the relationship between SGLT2 inhibitors and the occurrence of 9 types of non-infectious respiratory disorders [134]. Compared to placebo, SGLT-2is decreased the incidence of severe adverse events related to asthma. Another meta-analysis with a similar design concluded that SGLT-2is reduced the risk of asthma in comparison with GLP-1RAs and DPP-4is [131]. Despite that, the extremely low incidence of asthma outcomes in both groups (treatment and placebo) limits the validity of both meta-analyses.

Thiazolidinediones (TZDs) are another class of antidiabetic agent. They bind to the gamma isoform of the peroxisome proliferator-activated receptor (PPARγ) and reduce insulin resistance and ectopic fat accumulation [135,136]. So far, the results of studies regarding the anti-inflammatory effect of TZDs on asthma are controversial. Τroglitazone reduced IL-6 in cultured human airway smooth muscle cells, in a dose-dependent manner [137], but a systematic review concluded that TZDs had no significant effect on IL-6 levels [138]. A large observational retrospective study of diabetic Veterans who had a diagnosis of asthma and were taking oral antidiabetic agents [139], showed that exposure to thiazolidinediones was linked with a significant reduction in the risk of asthma exacerbation and oral steroid prescription. Utilization of angiotensin-converting enzyme inhibitors (ACE-I) and/or TZD was also linked with a lower risk for incident asthma in obese/overweight patients with diabetes and/or hypertension, in another retrospective observational longitudinal data analysis, who included 77,278 Veterans with incident asthma [140].

Conversely, in a 12-week, randomized, placebo-controlled, double-blind trial, no difference was observed in asthma control, exhaled nitric oxide, or lung function between treatment groups [141]. Interestingly, patients receiving pioglitazone gained significantly more weight than those receiving placebo [141]. New safety concerns regarding the risk of bladder cancer with pioglitazone led to the early discontinuation of the study [141]. Finally, a RCT of pioglitazone in severe asthma found no beneficial effect on asthma quality of life, as well as many untoward effects [142]. Thus, TDZs do not appear to hold for asthma.

The potential effects of current classes of hypoglycemic therapies in the pathophysiology of asthma are shown in Table 2.

## 6. Conclusions

Obesity, insulin resistance or glucose intolerance, dyslipidemia, and other key clinical features of metabolic dysfunction—typically manifested as metabolic syndrome—have been found to be possible risk factors for severe and uncontrolled asthma. More specifically, disorders of glucose metabolism, ranging from clinically silent insulin resistance through degrees of hyperglycemia defining prediabetes and diabetes, can precipitate changes in the lung consistent with asthma through pathways principally involving insulin excess. Recent experimental studies have shown that insulin resistance may contribute to an increase in systemic inflammation, modulation of immune function, effects on airway remodeling, promotion of airway smooth muscle (ASM) contractility and proliferation and increase of airway hyper-responsiveness. Evidence from observational studies also supports clinically relevant associations between glycemic dysregulation and asthma outcomes. In our opinion, there are four mechanisms of the effect of insulin resistance in asthma: (i) aging is associated with insulin resistance, which can lead to premature airway closure and airway damage, (ii) insulin directly contributes to airway dysfunction by causing airway inflammation through the activation of immunological and structural cells in the lungs, (iii) insulin can directly induce airway hyperresponsiveness by promoting the deposition of collagen fibroblasts in the airways, (iv) insulin is a pleiotropic hormone that affects endothelial cells in a variety of ways. When all the aforementioned are taken into consideration, a significant unmet therapeutic demand for drugs to ameliorate asthma management and control in patients with insulin resistance is revealed. A growing volume of epidemiologic evidence indicates that by focusing on the underlying glycemic dysregulation, specific types of hypoglycemic drugs may be able to treat both chronic diseases. To better understand the influence and the role of insulin resistance in asthma, more research is required, and novel antidiabetic drugs pertaining to asthma should be investigated. Expanding our understanding of the diverse range of biochemical mechanisms that result in an adult asthma diagnosis may also enable us to pinpoint important tactics to prevent insulin resistance, obesity, and asthma from emerging later in life.

## Figures and Tables

**Figure 1 biomedicines-12-00437-f001:**
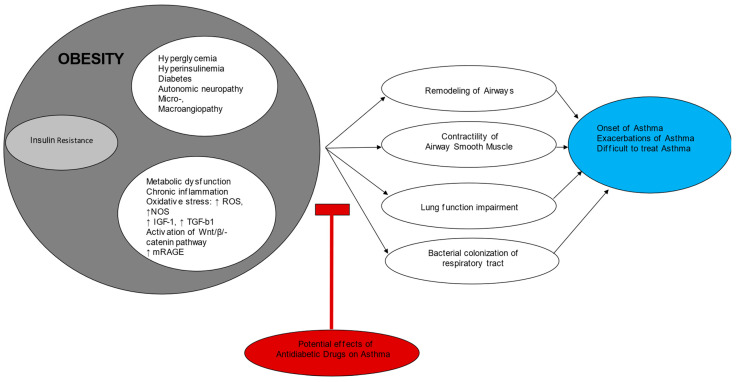
Insulin resistance and asthma. IGF-1: insulin-like growth factor, mRAGE: membrane receptor advanced glycation end products, NOS: nitric oxide species, ROS: reactive oxygen species, TGF-b1: transforming growth factor b1.

**Table 1 biomedicines-12-00437-t001:** Suggested mechanisms of insulin resistance in asthma.

**Enhanced Airway Remodeling**
Increased collagen deposition in the lungs
Increased epithelium to mucus transition
Increased fibrosis
Increased airway smooth muscle mass
Increased expression of TGF-β1
**Increased Airway Smooth Muscle (ASM) contractility**
Increased airway hyper-responsiveness
Loss of inhibitory M2 muscarinic receptor function
Enhanced vagally mediated bronchoconstriction
**Lung function impairment**
Reduced Forced Expiratory Volume in the 1st sec (FEV1)
Reduced Forced vital capacity (FVC)
Reduced Forced expiratory flow over the middle half of the FVC (FEF25–75%)
**Increase the risk of respiratory tract bacterial colonization**
**Release of pro-inflammatory mediators from adipose tissue**
Increased IL-6
Increased TNF-α
Increased Th2-inflammation

**Table 2 biomedicines-12-00437-t002:** Potential effects of current classes of antidiabetics in pathophysiology of asthma.

Antidiabetic Agent	Potential Beneficial Effects on Asthma	Putative Mechanisms
Metformin	Yes	Reverse lung tissue eosinophilic infiltrationProinflammatory cytokines ↓ Reactive Oxygen Species ↓ Nitric Oxide Species ↓ TNF-α ↓ Cell adhesion molecules ↓ Allergic eosinophilic inflammation
GLP-1	Yes	↓ Proinflammatory cytokines ↓ TNF-α ↓ receptors of advanced glycation end-products (RAGE) expression ↓ Bronchoconstriction ↓ T2 cytokines ↓ Mucus hypersecretion ↓ Airway hyperresponsiveness (AHR) ↓ IL-33 ↓ Contractile tone
Dipeptidyl peptidase-4 inhibitors (DPP-4is)	Uncertain	↓ Oxidative stress ↓ Lung fibrosis
Sodium-glucose cotransporter-2 inhibitors (SGLT-2)	Yes	↓ TNF receptor 1 ↓ IL-6 ↓ Matrix metalloproteinase 7 ↓ Fibronectin 1
Thiazolidinediones (TZD)	No	↓ IL-6 *
Sulfonylureas	Uncertain	

* A systematic review concluded that TZD does not significantly affect IL-6 levels. ↓: reduce.

## Data Availability

Data are available upon request.

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
