# Peer review of "Unraveling the Link between Ιnsulin Resistance and Bronchial Asthma"

_biomedicines, 2024, doi:10.3390/biomedicines12020437_

Round 1

Reviewer 1 Report

Comments and Suggestions for Authors

The authors present a review article on mechanisms of action and existing preclinical clinical and clinical data regarding the effect of insulin resistance in asthma.

 There are some points to be addressed, which will improve the quality of the paper:

1. The manuscript would benefit from a graphical representation of insulin resistance and asthma pathophysiology. In the graphical representation, please also clearly indicate which data were derived from human studies and which were from animal models.

2. Please correct typing errors:

-line 212 missing bracket (ASL)

-line 388 delete TD: “Very recently, Wu TD et al….”

-line 442 add SGLT and delete inhibitors (SGLT-2is): “…relationship between -2is inhibitors and…”

-line 454 replace “who” with “with”: “A large retrospective observational study of diabetic Veterans who asthma and were…”

3. In Table 2 replace “Suppress” in “Suppress receptors of…” with “↓” similar as in other mechanisms. In the same table add “s” in “DPP-4i”.

In the same table for consistency, use the same terminology as in the text for SGLT-is = sodium-glucose cotransporter-2 inhibitors.

Comments on the Quality of English Language

English language is fine. There are some typing errors that needs to be corrected as indicated in the Comments and Suggestions for Authors.

Author Response

Reviewer 1

Comments and Suggestions for Authors

The authors present a review article on mechanisms of action and existing preclinical clinical and clinical data regarding the effect of insulin resistance in asthma.

There are some points to be addressed, which will improve the quality of the paper:

  1. The manuscript would benefit from a graphical representation of insulin resistance and asthma pathophysiology. In the graphical representation, please also clearly indicate which data were derived from human studies and which were from animal models.

Answer: We thank the reviewer for this comment. A graphical representation of insulin resistance and asthma pathophysiology has been added accordingly.

  1. Please correct typing errors:

-line 212 missing bracket (ASL)

-line 388 delete TD: “Very recently, Wu TD et al….”

-line 442 add SGLT and delete inhibitors (SGLT-2is): “…relationship between -2is inhibitors and…”

-line 454 replace “who” with “with”: “A large retrospective observational study of diabetic Veterans who asthma and were…”

Answer: Spelling and typing errors have been corrected as per your suggestions.

  1. In Table 2 replace “Suppress” in “Suppress receptors of…” with “↓” similar as in other mechanisms. In the same table add “s” in “DPP-4i”.

Answer: It has been corrected as per your suggestions.

In the same table for consistency, use the same terminology as in the text for SGLT-is = sodium-glucose cotransporter-2 inhibitors.

Answer: It has been corrected as per your suggestions.

Reviewer 1

Comments and Suggestions for Authors

The authors present a review article on mechanisms of action and existing preclinical clinical and clinical data regarding the effect of insulin resistance in asthma.

There are some points to be addressed, which will improve the quality of the paper:

  1. The manuscript would benefit from a graphical representation of insulin resistance and asthma pathophysiology. In the graphical representation, please also clearly indicate which data were derived from human studies and which were from animal models.

Answer: We thank the reviewer for this comment. A graphical representation of insulin resistance and asthma pathophysiology has been added accordingly.

  1. Please correct typing errors:

-line 212 missing bracket (ASL)

-line 388 delete TD: “Very recently, Wu TD et al….”

-line 442 add SGLT and delete inhibitors (SGLT-2is): “…relationship between -2is inhibitors and…”

-line 454 replace “who” with “with”: “A large retrospective observational study of diabetic Veterans who asthma and were…”

Answer: Spelling and typing errors have been corrected as per your suggestions.

  1. In Table 2 replace “Suppress” in “Suppress receptors of…” with “↓” similar as in other mechanisms. In the same table add “s” in “DPP-4i”.

Answer: It has been corrected as per your suggestions.

In the same table for consistency, use the same terminology as in the text for SGLT-is = sodium-glucose cotransporter-2 inhibitors.

Answer: It has been corrected as per your suggestions.

Reviewer 2 Report

Comments and Suggestions for Authors
  1. The current article thoroughly explores the intricate relationship between insulin resistance, metabolic dysfunction, and asthma, providing a detailed review of the existing literature.
  2. The topic is highly relevant, given the rising prevalence of both asthma and metabolic syndromes like diabetes.
  3. The paper is well-structured, with clear subdivisions that guide the reader through different aspects of the topic.

I have some concerns which authors should address before accepting the paper for publication:

  1.  
  2.  
  3. - References are somewhat outdated. Authors should include more recent studies to ensure that the review reflects the latest findings in the field.
  4.  
  5. - A deeper discussion on the practical, clinical implications of these findings would enhance the paper’s applicability.
  6.  
  7. - The paper could benefit from incorporating a wider range of perspectives, including potential counterarguments or alternative explanations for the observed phenomena.

- I found several english errors throughout the manuscript. Please have a deep language revision. 

- An eye-catching figure depicting the Suggested mechanisms of insulin resistance in asthma would really increase the quality of the article.

Comments on the Quality of English Language

moderate revision needed 

Round 2

Reviewer 1 Report

Comments and Suggestions for Authors

Although the authors replied that they have added “a graphical representation of insulin resistance and asthma pathophysiology”, I can not find the figure in the revised manuscript.

Please add the figure.

Author Response

Although the authors replied that they have added “a graphical representation of insulin resistance and asthma pathophysiology”, I can not find the figure in the revised manuscript. Please add the figure.

Answer: The figure isn incorporated now to the manuscipt (it was uploaded as a separated file)

Reviewer 2 Report

Comments and Suggestions for Authors

Ok to accept now

Author Response

Ok to accept now

Answer: We thank the Reviewer for his/her comment.

Round 3

Reviewer 1 Report

Comments and Suggestions for Authors

The authors have addressed all the questions, and the manuscript is improved and acceptable for publication.